A global meta-analytic contrast of cushion-plant effects on plants and on arthropods

Liczner Amanda R.
Lortie Christopher J. lortie@yorku.ca
Department of Biology, York University , Toronto, Ontario , Canada
Sanders Nathan
Electronic publication date: 2014 Feb 27
Publication date: 2014
Volume: 2
Electronic Location ID: e265
Received 2013 Nov 26; Accepted 2014 Jan 19
Copyright: © 2014 Liczner et al.
Copyright year: 2014
Copyright holder: Liczner et al.
License: This is an open access article distributed under the terms of the Creative Commons Attribution License, which permits unrestricted use, distribution, and reproduction in any medium, provided the original author and source are credited.
License URL: https://creativecommons.org/licenses/by/3.0/

Keywords: Arthropods, Facilitation, Meta-analyses, Nurse plants, Cushion plants

Funding: The United States Department of the Interior Bureau of Land Management NSERC DG NSERC Canadian Pollination Initiative (CANPOLIN) This research was funded by The United States Department of the Interior Bureau of Land Management, an NSERC DG, and an NSERC Canadian Pollination Initiative (CANPOLIN) grant to CJL. This is publication #109 of NSERC-CANPOLIN. The funders had no role in study design, data collection and analysis, decision to publish, or preparation of the manuscript.

==============================
Nurse plant facilitation is a commonly reported plant–plant interaction and is an important factor influencing community structure in stressful environments. Cushion plants are an example of alpine nurse plants that modify microclimatic conditions within their canopies to create favourable environments for other plants. In this meta-analysis, the facilitative effects of cushion plants was expanded from previous syntheses of the topic and the relative strength of facilitation for other plants and for arthropods were compared globally.The abundance, diversity, and species presence/absence effect size estimates were tested as plant responses to nurse plants and a composite measure was tested for arthropods. The strength of facilitation was on average three times greater for arthropods relative to all plant responses to cushions. Plant species presence, i.e., frequency of occurrence, was not enhanced by nurse-plants. Cushion plants nonetheless acted as nurse plants for both plants and arthropods in most alpine contexts globally, and although responses by other plant species currently dominate the facilitation literature, preliminary synthesis of the evidence suggests that the potential impacts of nurses may be even greater for other trophic levels.

Introduction

Facilitation is a positive, non-trophic interaction that benefits at least one species (Callaway, 1995; Bruno, Stachowicz & Bertness, 2003). This interaction tends to occur in high-stress environments such as deserts (Holzapfel & Mahall, 1999) or arctic and alpine ecosystems (Antonsson, Bjork & Molau, 2009). The importance of facilitation tends to increase with environmental stress (Choler, Michalet & Callaway, 2001; Brooker et al., 2008; le Roux & McGeoch, 2010). A commonly used tool to examine plant facilitation in stressful environments is the use of nurse plants. Nurse plants modify microclimatic conditions of stressful environments within their canopies and thus may increase species richness (Nunez, Aizen & Ezcurra, 1999; Arroyo et al., 2003; Badano & Marquet, 2009), abundance (Cavieres et al., 2002; Badano et al., 2007; Sklenar, 2009), diversity (Badano & Marquet, 2009; Butterfield et al., 2013), and species survival (Cavieres et al., 2007; Badano et al., 2007; Cavieres, Quiroz & Molina-Montenegro, 2008). Less commonly, nurse plants can also increase seedling tolerance to herbivory (Acuna-Rodriguez, Cavieres & Gianoli, 2006). Cushion plants are nurse plants that grow in alpine, subalpine, arctic, and subarctic ecosystems. The physiology of cushion plants, including their low height and compact form, makes them well adapted to stressful alpine environments. It also allows them to alter microclimatic conditions within their canopies (Cavieres et al., 2006). The canopy traps heat providing a warmer microclimate for other plants to grow in (Arroyo et al., 2003; Molenda, Reid & Lortie, 2012), increases soil water content by retaining moisture (Cavieres et al., 2007; Schoeb, Butterfield & Pugnaire, 2012; Anthelme et al., 2012), reduces wind (Cavieres et al., 2007; le Roux & McGeoch, 2010), and increases litter accumulation which contributes to increased soil nutrients (Cavieres, Quiroz & Molina-Montenegro, 2008; Schoeb, Butterfield & Pugnaire, 2012; Anthelme et al., 2012). Consequently, cushion plants are an excellent set of species to explore positive interactions in the alpine, particularly for impacts on other plant species.

Most cushion plant facilitation studies have focused on the facilitation of other plants with few examples of effects on arthropods (but see Molina-Montenegro, Badano & Cavieres, 2006; Sieber et al., 2011; Lortie & Reid, 2012; Molenda, Reid & Lortie, 2012). Accordingly, reviews of cushion plant facilitation have also focused on plants (Arredondo-Nunez, Badano & Bustamante, 2009; Anthelme & Dangles, 2012; Reid, Lamarque & Lortie, 2010). For instance, Anthelme & Dangles (2012) examined plant–plant interactions in tropical alpine environments and compared them to other alpine environments. They found that cushions have a similar facilitative effect in tropical alpine environments to other alpine environments in that cushions modified microclimatic conditions within their canopies and similarly facilitated other plant species. A review by Reid, Lamarque & Lortie (2010) compared publications on nurse-plant shrubs to cushion plants and found that although there are fewer studies using cushion plants, these nurses have many of the same effects as shrubs in terms of modifying microclimatic conditions and enhancing plant species diversity. Cushions are thus an ideal model in many respects to study the effects of facilitation on diversity in alpine or arctic ecosystems. However, these two reviews summarized cushion plant effects on understory species across studies, and did not quantitatively assess these effects in terms of richness, abundance, survival etc.

A pioneering meta-analysis by Arredondo-Nunez, Badano & Bustamante (2009) quantitatively examined the effect of cushion plants on plant species presence at high and low stress and concluded that facilitation increased with environmental stress. This meta-analysis was very successful in quantitatively demonstrating the facilitative effects of cushions on other plant species. However, the literature available for synthesis at that time only examined plant–plant interactions in the Southern Andes, and only tested plant species presence/absences a response variable in sufficient numbers (a total of 9 studies). Hence, there is need for a quantitative synthesis update for these specific forms of nurse plants.

In this meta-analysis, the effect of cushion plants will first be extended to assess effects on other plant species globally by comparing the following three potential plant responses to nurses: abundance, diversity, and presence/absence. The synthesis of facilitation literature will be further extended by contrasting the responses of other plant species to nurses with the responses of arthropods to nurses. These are critical extensions to the facilitation literature in general because nurse plants may be foundation species for many trophic levels – not just other plants, and identifying the appropriate responses to nurse plants may have important implications for population and community dynamics of alpine plant populations. The following questions will be addressed in this meta-analysis. (1) Is there significant evidence that cushion plants facilitate plant abundance, diversity, and presence/absence globally? (2) Is there significant evidence that cushion plants also facilitate arthropods? (3) Does the strength of evidence associated with facilitation of plants and arthropods by cushion plants differ? Plants and arthropod responses to cushions can be contrasted herein because the same effect size estimate is calculated and both involve the same field of methodologies, i.e., contrasts of measures associated with cushion and open microsites. This satisfies the best practices recommended for such meta-analytical contrasts (Moayyedi, 2004; Borenstein et al., 2009; Jennions, Lortie & Koricheva, 2013; Cote & Jennions, 2013) when the efficacy of treatment is evaluated at larger scales.

Methods

Study selection process

A search was conducted using ISI Web of Knowledge for articles associated with cushion plant facilitation. Three separate searches were performed in July 2013 on this topic and resulted in 613 articles (Table 1, search terms listed). These searches were refined in three stages with increasing specificity in the inclusion criteria applied. The first stage limited articles to English language publications and to the following Web of Knowledge search categories: plant science, ecology, environmental sciences, geography physical, environmental studies, biodiversity conservation, evolutionary biology, horticulture, entomology, biology, and mycology (Table 1, 432 articles remained). In the second stage of refinement, duplicate articles were removed, and all publications were screened to determine if the study examined facilitation (retention of 52 articles). Only two taxa were reported in this set of publications, plants, and arthropods. The third stage in the workflow inspected all studies for useable/extractable data and then sorted these publications by response variables, i.e., abundance, diversity, and presence, and by plant or arthropod species. This final refinement generated 16 studies for a total of 673 unique experimental contrasts of nurse-plant cushion effects in the field (Table 2, Nstudy = 13 plant responses, Nstudy = 2 arthropods, and Nstudy = 1 examined both taxa). A PRISMA flow diagram was generated (Moher et al., 2009) outlining the publication selection process (Fig. 1).

Figure 1 PRISMA diagram describing the search protocol used for the meta-analysis.

PRISMA flow diagram depicting the search protocol and workflow in determining the effective population of studies for meta-analysis.

Table 1 Search terms used to select studies.

The search terms used in defining the scope of studies used in this meta-analyses of nurse-plant cushions on other plant species and arthropods. Asterisks were included in the search terms as a Boolean search strategy to identify word variations. Web of Knowledge was the tool used to secure the population of studies. Each workflow step of literature screening is described in details in the methods, but in short, step 1 — all studies, step 2 — duplicates removed and reported facilitation, and step 3 — useable data reported and sorted by response and taxa.

Workflow	Search terms	Ninitial	N step1	N step2	N step3	
1	Cushion plant OR nurse plant AND facilitat* AND alpine OR arctic OR subarctic	30	30	27	13	
2	Cushion plant OR nurse plant AND faciliatat*	54	53	12	2	
3	Cushion plant	529	349	13	1	

Table 2 Article selection criteria for inclusion in the meta-analysis.

A summary of all articles included in the meta-analysis of nurse-plant cushions on plants and arthropods. Details of data extraction listed in detail in the methods (Nstudies = 16, nplants = 662, narthropods = 11).

Authors	Location	Elevation (m.a.s.l.)	Cushion species	Taxa	Response variable	
(Anthelme et al., 2012)	00°28′ S, 78°09′ W	4400, 4550, 4700	Azorella aretioides	Plants	Diversity, presence	
(Arroyo et al., 2003)	50°48′ S, 73°10′ W	700, 900	Azorella monantha	Plants	Presence	
(Badano et al., 2007)	33°S, 70°W	3200, 3400, 3600	Azorella monantha	Plants	Abundance	
(Cavieres et al., 2002)	50°48′ S, 73°10′ W	700, 900	Bolax gummifera	Plants	Abundance, diversity, presence	
(Cavieres et al., 2006)	33°20′ S, 70°16′ W	2800, 3200	Laretia acaulis	Plants	Diversity, presence	
(Cavieres, Quiroz & Molina-Montenegro, 2008)	33°20′ S, 70°16′ W	3200	Laretia acaulis, Azorella monantha	Plants	Abundance, presence	
(Cavieres & Badano, 2009)		1900, 1600, 1900, 3200,
3600, 4000, 4300	Pycnophyllum bryoides,
Adesmia suvterranea,
Azorella madreporica,
Laretia acaulis,
Oreopolus glacialis,
Discaria nana,
Mulinum leptacapthum,
Azorella monantha,
Bolax gummifera	Plants	Diversity	
(de Bello et al., 2011)	33°05′ N, 78°27′ E	5900	Thylacospermum caespitosum	Plants	Presence	
(Dvorsky et al., 2013)	34°45′ N, 77°35′ E	4840, 5000, 5100, 5300,
5600, 5750, 5850	Tylacospermum caespitosum	Plants	Diversity, presence	
(le Roux & McGeoch, 2010)	46°54′ S, 37°45′ E	89, 97, 102	Azorella selago	Plants	Abundance	
(Molenda, Reid & Lortie, 2012)	50°15′ N, 122°16′ W	2160	Silene acaulis	Plants and arthropods	Abundance, diversity	
(Molina-Montenegro, Badano & Cavieres, 2006)	33°20′ S, 70°16′ W	3200	Laretia acaulis,
Azorella monantha	Arthropods	Abundance	
(Quiroz, Badano & Cavieres, 2009)	33°20′ S, 70°16′ W	3200, 3580	Azorella madreporica	Plants	Abundance, diversity, presence	
(Schoeb, Butterfield & Pugnaire, 2012)	37°05′ N, 03°23′ W	3240	Arenaria tetraquetra	Plants	Abundance, diversity	
(Sieber et al., 2011)	46°31′ N, 09°43′ W	3000	Eritrichium nanum	Arthropods	Presence	
(Yang et al., 2010)	28°20′ N, 99°05′ E	4500, 4700	Arenaria polytrichoides	Plants	Presence	

Data collection and analyses

Data for abundance, diversity and/or presence of plant and/or arthropod species were extracted from tables, figures, or by contacting authors directly when not reported. All studies excepting one included in the meta-analysis were observational (Table 2). To compare results across studies, the Relative Interaction Index (RII) effect size estimate was calculated as RII = (Bw−Bo)/(Bw + Bo) where Bw is the value of species within the cushion, and Bo is the value of species without the cushion (Armas, Ordiales & Pugnaire, 2004). RII ranges from +1 to −1 with positive values indicating facilitation, negative values indicating competition, and values not significantly different from zero indicating neutral/no effects (Armas, Ordiales & Pugnaire, 2004). Sets of meta-analytic contrasts were used to compare the nurse effect of cushions on plants and to arthropods. The effect of cushions was determined by comparing plant and arthropod responses within the cushion canopy to adjacent open areas identical to the field methodology used to assess plant–plant interaction in most facilitation studies (Brooker et al., 2008). These nurse plant-open pairs were extracted from each study and used for each meta-analytic contrast resulting in 662 pairs for plants and 11 for arthropods. Pairs were first coded as a unique replicate/instance based on study number, cushion species, elevation, and response variable reported within the study (i.e., abundance, diversity, or species presence). However, to be very conservative, we chose not to model each field instance as fully independent in our analyses. The mean RII values were calculated within each publication for independent tests only, i.e., tested a different cushion species or a different elevation, for a total of 63 unique study cases for plants and 5 tests for arthropods. We first tested whether abundance, diversity, and presence differed between plants on average. Next, we compared the composite measure of all responses between plants and arthropods. Diversity data included raw species richness and Shannon-Weiner diversity indices. Both meta-analyses were modeled as categorical random effects. Heterogeneity tests (Q) were conducted to determine if the effect sizes calculated in each meta-analysis were significantly different (Rosenberg, Adams & Gurevitch, 2000). To determine if the effect size was significantly different from zero and therefore significantly different from a neutral effect, bias corrected confidence intervals were calculated. An effect size was significantly different from zero if the confidence interval does not overlap zero (Cote & Jennions, 2013). In order to explore bias, Rosenthal’s fail-safe analyses were conducted for each meta-analysis. To determine if the Rosenthal value for each meta-analysis is within the acceptable range, we applied the bias rule of X = 5k + 10 where X = the Rosenthal value and k is the number of studies (Moller & Jennions, 2001). An acceptable Rosenthal value for plants would be greater than 80 whilst for arthropods it would be greater than 25. If the Rosenthal value of the meta-analysis is greater than these values, then the results are generally considered robust (Moller & Jennions, 2001). All univariate meta-analyses were conducted using Metawin 2.1 (Rosenberg, Adams & Gurevitch, 2000).

Results

Plant abundance was the most strongly facilitated response variable enhanced by cushions, and it was significantly different from the other responses (Fig. 2, different from 0 and non-overlapping confidence intervals with either alternative response mean RIIabundance = 0.434 ± 0.144, mean RIIdiversity = 0.130 ± 0.081, mean RIIpresence = 0.095 ± 0.166). Plant species diversity was also enhanced by cushions whilst the presence plant response variable was not significantly different from zero (Fig. 2). Heterogeneity between groups was significantly different (Qbetween = 11.7, df = 2, p = 0.01) with presence plant response having the highest levels of within group variation (presence variancepooled within group = 0.13). The Rosenthal value for this meta-analytic comparison is 381 indicating robust results.

Figure 2 Mean RII values for the effect of cushion plants on the abundance, diversity, and presence of other plant species.

The mean RII values for the effect of alpine cushion plants on the abundance, diversity, and species presence for other plants. Presence refers to presence/absence responses via associational pattern analyses in this literature. The bias-corrected 95% confidence intervals are shown.

Cushion plants facilitated both plants and arthropods (Fig. 3, i.e., grand mean significantly different from zero and positive grand mean = 0.278 ± 0.082). The facilitative effect of cushion plants was significantly greater for arthropods compared to plants with arthropods having a RII value more than 3.5 times greater than plants (Fig. 2, mean RIIplants = 0.226 ± 0.079, mean RIIarthropods = 0.830 ± 0.041). Heterogeneity between groups was not significantly different (Qbetween = 3.3, df = 1, p = 0.08) in spite of unequal sample sizes. The Rosenthal value was 461.8, and this is 5 times greater than the threshold of 80 suggesting robust results.

Figure 3 Composite mean RII values for plants, arthropods and the overall grand mean.

A contrast of the composite mean RII values for plants and arthropods. The overall or grand mean is the mean RII value for both plants and arthropods. The bias-corrected 95% confidence intervals are shown.

Discussion

There is significant evidence that nurse-plant species function as foundation species in relatively stressful ecosystems (Cavieres & Badano, 2009; Butterfield, 2009). Particularly in arid systems, shrub nurse plants have been shown to positively influence many aspects of plant community structure (Maestre, Valladares & Reynolds, 2005), and this has been linked to restoration via synthesis, i.e., meta-analysis (Gómez-Aparicio, 2009). However, in the alpine, the research is not as extensive but also suggests that cushion plants can serve as foundation species with strong effects in driving the frequency of occurrence of plant species within these communities (Arredondo-Nunez, Badano & Bustamante, 2009). To update and extend this previous synthesis, we conducted a meta-analysis on the current research examining cushions to include other plant responses, other alpine regions globally, and to compare to effects on arthropods. Similar to the previous syntheses of nurse plants in general, cushion plants facilitate other plant species and arthropods and are thus likely a foundational species. Consequently, we propose that these species are an excellent model organism available to ecologists to explore community dynamics and change in many alpine ecosystems.

There were several novel and sometimes contradictory findings in this synthesis effort relative to previous reviews. In this meta-analysis, the abundance and diversity of plant species was facilitated by cushion plants. This is a novel extension to the previous synthesis by Arredondo-Nunez, Badano & Bustamante (2009) wherein only frequency of occurrence, or as we termed here presence, was examined. Increases in diversity and total abundance of plant species within the cushion understory is not a surprising result given the above described mechanisms of abiotic stress amelioration. Shelter in the alpine is a commonly assumed mechanism of facilitation for plants (Carlsson & Callaghan, 1991; Cavieres et al., 2002; Cavieres & Sierra-Almeida, 2012). There is accumulating support that cushions can enhance species richness in the alpine through higher rates of addition/retention of species at the community level (Cavieres & Badano, 2009). This retention by cushion plants has been shown to extend to reduced loss of phylogenetic diversity compared to adjacent open areas in the alpine globally (Butterfield et al., 2013). Even more broadly, facilitation can enhance diversity in many other ecosystems (Badano & Marquet, 2009; Butterfield et al., 2013; McIntire & Fajardo, in press). However, all species may not equally benefit from nurse plants in alpine systems, and there are also instances of negative association of other species with cushions (Fajardo, Quiroz & Cavieres, 2008) or different sets of species differentially associating with cushions (Cavieres & Badano, 2009; Arredondo-Nunez, Badano & Bustamante, 2009). This synthesis thus contradicted the previous synthesis of this topic (Arredondo-Nunez, Badano & Bustamante, 2009) in that the presence, or frequency of occurrence, was not facilitated as was formerly detected. This difference is likely due to several factors ranging from ecological to statistical. The current meta-analysis included studies from a variety of alpine ecosystems because the cushion plant literature has expanded in number and geographic scope since the former synthesis. This necessarily introduces greater heterogeneity in the potential responses of plant communities to cushions because very different alpine communities were sampled that likely differ in stability (Butterfield, 2009), net interactions (Callaway et al., 2002), and climate (Kikvidze et al., 2011) to name a few important ecological considerations. Importantly, significant statistical heterogeneity was detected for the presence plant response variable unlike the other responses tested suggesting that this measure of community structure may be more sensitive the local ecological context versus regional drivers of change (Ricklefs, 2008). The inconsistency between this study and the meta-analysis conducted by Arredondo-Nunez, Badano & Bustamante (2009) is also due to purely statistical reasons because the scope of inference differed. Herein, we fit random-effects statistical models as we sought to describe global patterns whilst the former meta-analysis, quite appropriately, used a fixed-effects model because they were describing a set of studies all from within the same region, the Southern Andes. Random effects models estimate variance less conservatively (Jennions, Lortie & Koricheva, 2013), and we would thus expect that heterogeneity would be greater in some instances. In summary, cushion plants have the capacity to shape many aspects of plant community structure in the alpine, but research gaps associated with species specificity, scale, and the sensitivity of different community-level responses to nurses can be further developed.

Although the facilitation of arthropods is an emerging field of research, arthropods were facilitated by cushion nurse plants in the alpine in this limited set of studies conducted to date. Interestingly, the strength of facilitation was significantly greater for arthropods relative to the benefits accrued by other plant species. There are several explanations for this general finding. Microclimatic modifications made by cushion plants may benefit arthropods even more extensively than plants given their mobility and foraging behaviour. The canopy of cushions provides a warmer and more stable microclimate (Cavieres et al., 2002; Arroyo et al., 2003; Molenda, Reid & Lortie, 2012). This may allow more arthropods to function and thermoregulate relative to colder conditions outside cushions (Molina-Montenegro, Badano & Cavieres, 2006). If sets of arthropods seek refuge within cushions, then the availability of prey may also be greater within cushions thereby concentrating resources for other species (Lortie & Reid, 2012). Cushion plants also increase plant abundance and diversity when compared to open areas in many instances (finding in this synthesis and broadly reviewed in McIntire & Fajardo (in press)). This can provide arthropods with a more diverse range of resources and niches in general (Molenda, Reid & Lortie, 2012) particularly for life-stages associated with colonization (Mysterud et al., 2010). Finally, pollinators have been shown to benefit from cushions as they provide an increased availability of flowers (Reid & Lortie, 2012). Hence, cushion plants likely have direct and indirect effects on arthropod community dynamics related to both microclimate and to the other plant and arthropod species present. The evidence to date strongly suggests that cushion nurse plant research should now include and address multi-trophic perspectives (McIntire & Fajardo, in press; Van Der Putten, 2009; Ferenc, Liu & Mike, 2009). In addition, decoupling direct and indirect interactions of cushion plants and understory plant species is another important area of research. Biodiversity changes in the alpine will be unavoidable given a changing climate and will not be restricted to plant species. Therefore, understanding interactions that structure the greater community will be important in determining the consequences of a rapidly changing climate in the alpine.

Supplemental Information

Supplemental Information 1 PRISMA checklist

PRISMA checklist showing the page number of each of the PRISMA criteria.

Click here for additional data file.

Additional Information and Declarations

Competing Interests

Author Contributions

Christopher J. Lortie is an Academic Editor for PeerJ.

Amanda R. Liczner analyzed the data, wrote the paper, prepared figures and/or tables, reviewed drafts of the paper.

Christopher J. Lortie analyzed the data, contributed reagents/materials/analysis tools, wrote the paper, prepared figures and/or tables, reviewed drafts of the paper.

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
