# Peer review of "A global meta-analytic contrast of cushion-plant effects on plants and on arthropods"

_PeerJ, doi:10.7717/peerj.265_

## Round 0.1 · original submission · Major Revisions

Both reviewers like the idea of the paper and think the topic is interesting. However, both raise substantial concerns.

Reviewer 2 mentions the lack of "striking implications." I"m not so concerned about that. I am concerned, however, that the analysis is robust enough to allow you to make any substantial claims. Both of the reviewers and I wondered about whether the fundamental comparison between arthropods and plants can be made and whether the conclusions can be supported when there are only 3 studies to be included in the meta-analysis.

Both reviewers also point out some areas of the manuscript that need to be clarified or elaborated on. I will in all likelihood send your revision back to these same two reviewers. I think you have your work cut out for you if you're going to mollify them.

Reviewer 1 ·

Basic reporting

This paper explores global patterns of cushion plants role in acting as nurse plants (facilitators) for other plants, and compares this with the facilitating strength for arthropods. I think the paper is overall well written and explores a very interesting topic. However, I have some comments and questions. Please find them listed below.

1) I think the introduction would benefit greatly from a section that introducing the arthropod story, and why the comparison between arthropods and plants is interesting and important. It is a novel, and a major, part of this paper. And, for readers that are used to only the plant literature I think such an introduction would be very helpful.

2) Figure 2 is lacking (and Figure 3 has been inserted twice by mistake).

Experimental design

1) The question(s) of the paper needs to be described better. In connection with this please see my comment above where I think a clear explanation/justification for the comparison between nurse plant effects on plants versus arthropods is needed for the paper to cohere to the point “The submission should clearly define the research question, which must be relevant and meaningful”.

2) Lines 117-118: Please expand on this text. As it reads now I do not understand how the authors end up with these numbers.

Validity of the findings

1) Given the large difference in the number of studies on plants and arthropods compared here (mainly that the one of the two groups contains three studies only) is it reasonable to draw strong conclusions regarding the variation in strength of facilitation/nurse plant effects on plants and arthropods?

Additional comments

Minor comments:

Line 46: insert “space” between “2010).Cushions”
Line 55: insert “space” between “2010).For instance”
Lines 64-65: In the first part of the sentence please explain in just a little more detail what the impact of cushion plants were on. There seems to be something missing in the second part of the sentence “…within-study strength of evidence of cushion plants”. Please expand on this just a little further.
Lines 132-133: Repetition of “the results are”.
There is a change in tense (past to present) in the Results text.
Line 183: insert space between “systems,and”
Line 220: remove “-“ before particularly?

Reviewer 2 ·

Basic reporting

I feel that the article is not satisfactorily edited from a grammatical and stylistic point of view. In addition, the fact that the same figure was provided twice (and one was apparently left out) shows that the manuscript was not very thoroughly checked before submitting. See below for some specific comments on word choice and sentence structure. Another general comment is that it was not clear exactly what type of studies (i.e., experimental or observational) were being compared in the meta-analysis. It would be informative to the reader to describe the general design of the analyzed studies in more detail.

Experimental design

To the best of my knowledge, the authors did a satisfactory job with the meta-analysis. However, the main concern I have is that while there are a number of semi-independent experimental contrasts within each of the published studies they analyzed, the number of publications (fully independent studies) is much too small, especially for arthropods (3). In addition, I feel strongly that meta-analyses that only calculate grand means are not as informative or useful as meta-analyses that exhaustively search for ways to explain the heterogeneity of effect sizes among studies. The only factor given to explain any heterogeneity in effect size was taxon (plant vs. arthropod), although the studies varied greatly in species of cushion plant, elevation, latitude, etc. The other issue that I feel is important to consider in meta-analyses is whether the various subgroup means of effect sizes are truly comparable. The authors calculated a grand mean effect size that lumps together the effect of cushion plants both on plants and on arthropods. This might not be a valid comparison, although I would be willing to concede the point if there were more studies to compare.

Validity of the findings

The topic the authors chose is a good one and is relevant to a large body of theory in plant ecology dealing with interactions among plants under conditions of environmental stress and along environmental gradients. Papers on facilitation are typically a welcome addition. However, I feel that the presentation is flawed and the sample size is too small to allow much inference to be drawn. Furthermore, the authors did not sufficiently develop implications of their findings in the discussion section. For all these reasons, I recommend that the manuscript be rejected.

Additional comments

The meta-analysis is technically sound and on a topic that is interesting. However, I felt that given the small sample size and the lack of striking implications that would advance our knowledge of this topic, the manuscript should be rejected. I am providing some line-by-line comments.

Line-by-line comments
34: “benefit other plants by increasing species richness”: phrasing is odd. Nurse plants benefit other plants by changing the microclimate or whatever other mechanism, and increased richness, survival, etc. may result from that.
42: “allowing them to alter microclimatic conditions”: What is the mechanism? How/why are the conditions altered?
44: “retains moisture by increasing soil water content”: again, the phrasing is odd. What is the mechanism by which moisture is retained and soil water content increases?
48: “particularly for other impacts on other plant species”: confusing wording.
71: “only tested. . . in sufficient numbers”: wording is confusing; I do not understand this phrase.
72: “quantitative synthesis update”: strange-sounding phrase.
81: “informs implications”: strange-sounding phrase.
118: “The first meta-analysis tested”: I think it would be better phrasing to say that you as the researcher tested something using a meta-analysis.
137: “was the most positive response variable enhanced”: confusing wording.
141: “significantly different” should read “significantly greater than zero.”
146: “are facilitating” should read “facilitate”
154: “at many points along environmental gradients”: I think this refers to both high- and low-temperature environments, but it is confusing as worded here.
156: “synthesis of shrub nurse plants have been shown”: bad wording.
165: “likely foundational in many respects”: vague phrase without much content.
166: “excellent tool. . . to explore community dynamics”: also somewhat vague.
167: “There were. . . reviews.”: This sentence could be made more specific.
170: “facilitated the abundance and diversity”: confusing wording.
173: “in many respects”: Which ones?
179: “reduced loss”: Not clear what this is compared to.
208: “Albeit an emerging field”: confusing wording.
208: “clearly”: Is this justified given the small number of studies?
218: At what scale is abundance and diversity increased?
222: “by increasing the availability of flowers in general”: confusing wording.
225: “highly viable”: word choice?
227: The last sentence of this paragraph seems somewhat out of context.
Figure 1: I am not sure this figure is necessary, as it is not a result of your study as such, and essentially restates information already found in the methods. The same applies to Table 1.
Figure 2-3: One of the figures was left out, since these are both the same figure.

---

## Round 0.2 · accepted · Accept

Nice work. This is a more clear, better manuscript for your efforts.